# Land Cover Classification from Hyperspectral Images via Weighted Spatial-Spectral Kernel Collaborative Representation with Tikhonov Regularization

**Rongchao Yang** 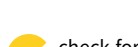**, Beilei Fan, Ren Wei, Yuting Wang and Qingbo Zhou ***

Agricultural Information Institute, Chinese Academy of Agricultural Sciences, Beijing 100081, China; yangrongchao@caas.cn (R.Y.); fanbeilei@caas.cn (B.F.); weiren@caas.cn (R.W.); wangyuting@caas.cn (Y.W.)
* Correspondence: zhouqingbo@caas.cn

**Abstract:** Precise and timely classification of land cover types plays an important role in land resources planning and management. In this paper, nine kinds of land cover types in the acquired hyperspectral scene are classified based on the kernel collaborative representation method. To reduce the spectral shift caused by adjacency effect when mining the spatial-spectral features, a correlation coefficient-weighted spatial filtering operation is proposed in this paper. Additionally, by introducing this operation into the kernel collaborative representation method with Tikhonov regularization (KCRT) and discriminative KCRT (DKCRT) method, respectively, the weighted spatial-spectral KCRT (WSSKCRT) and weighted spatial-spectral DKCRT (WSSDKCRT) methods are constructed for land cover classification. Furthermore, aiming at the problem of difficulty of pixel labeling in hyperspectral images, this paper attempts to establish an effective land cover classification model in the case of small-size labeled samples. The proposed WSSKCRT and WSSDKCRT methods are compared with four methods, i.e., KCRT, DKCRT, KCRT with composite kernel (KCRT-CK), and joint DKCRT (JDKCRT). The experimental results show that the proposed WSSKCRT method achieves the best classification performance, and WSSKCRT and WSSDKCRT outperform KCRT-CK and JDKCRT, respectively, obtaining the OA over 94% with only 540 labeled training samples, which indicates that the proposed weighted spatial filtering operation can effectively alleviate the spectral shift caused by adjacency effect, and it can effectively classify land cover types under the situation of small-size labeled samples.

**Keywords:** land cover classification; hyperspectral images; kernel collaborative representation; weighted spatial-spectral features

## 1. Introduction

The accurate classification of land cover types is the key and important foundation for land cover mapping. Precisely and timely updating land cover mapping information can provide important theoretical basis for decision-making of land resource planning and management, environmental protection, precision agriculture, landscape pattern analysis, and so on [1–3]. Although the traditional land cover information collection method based on field survey can provide accurate land cover details, it costs a lot of manpower and time, and it cannot be carried out under some environmental conditions [1]. With the rapid development of sensor technology, various remote sensing detection technologies are emerging. Additionally, remote sensing technology has become one of the most important means of land cover mapping, because it can efficiently and contactlessly obtain ground object information on a large scale [4,5]. In the past few years, researchers have utilized various remote sensing data to classify and map land cover types and achieved satisfactory results, such as satellite or airborne RGB images [6,7], multispectral images [8,9], hyperspectral images [10,11], synthetic aperture radar [12,13] and multi-source remote sensing

images composed of the above data sources [14,15]. Among these remote sensing technologies, hyperspectral images can provide abundant spectral and spatial information for land cover objects, due to containing hundreds of narrow and continuous spectral bands [16,17]. Therefore, it has attracted extensive attention from scholars in the research of land cover classification and mapping.

Land cover classification using hyperspectral images is essentially to assign predefined label information to each pixel in hyperspectral images. In recent years, collaborative representation classification (CRC) model has been widely used in hyperspectral images for land cover classification and mapping, which was first developed for face recognition [18]. On the one hand, CRC utilizes a dictionary composed of all labeled training samples to linearly represent each test sample without considering any prior distribution of samples [19]. On the other hand, CRC employs an $\ell_2$-norm minimization-derived closed-form solution to solve the dictionary representation coefficient for each test sample, which possesses higher operation efficiency and better classification performance than that of the sparse representation classification (SRC) model [20].

To improve the classification performance of the collaborative representation (CR)-based model in hyperspectral images, Li et al. introduced a distance-weighted Tikhonov regularization into the original nearest-subspace classification (NSC, also called pre-partitioning CR model) and CRC (also called post-partitioning CR model), defined as NRS [21] and CRT [22], respectively. However, in the real hyperspectral images, the sample data are often presented in nonlinear structure, and the linear representation of CR models cannot fully represent the nonlinear structure of samples [19]. To solve this problem, many researchers utilize the kernel trick to project the sample data into a nonlinear high-dimensional feature space, where the separability of samples is improved [19,22–24]. For example, Li et al. incorporated the Gaussian radial basis function (RBF) kernel into the CRT method, which was denoted as KCRT and effectively improved the separability of land cover types in hyperspectral images [22]. Based on this work, Ma et al. proposed a discriminative kernel collaborative representation method with Tikhonov regularization (DKCRT) for land cover classification, which was able to make the kernel collaborative representation of different land cover types to be more discriminative in hyperspectral images [23].

Furthermore, many studies have shown that the combination of spatial and spectral features can effectively improve the performance of CR models for land cover classification in hyperspectral images [22–27]. Among them, a spatial filtering operation is a frequently used method to mine spatial-spectral features by directly averaging all the pixels (central pixels) and its corresponding spatial neighborhood pixels in hyperspectral images, such as KCRT with composite kernel (KCRT-CK) [22] and joint DKCRT (JDKCRT) [23]. In hyperspectral images, although each central pixel and its corresponding adjacent pixels belong to the same class in a high probability, it usually includes some pixels of different classes from the central pixel in its adjacent pixels. Moreover, when acquiring the spectral information of ground objects in the same scene, the hyperspectral sensor collects the direct reflection power from the central pixel and the indirect diffuse reflection powers from its adjacent pixels at the same time. Therefore, the spectral curve of the central pixel produces spectral shift affected by these adjacent pixels, which is called adjacency effect [28]. If the spatial adjacent pixels are averaged directly, the reconstructed central pixel (i.e., spatial-spectral features) will contain a large amount of noise caused by spectral shift, which affects the performance of CR models for land cover classification. For each central pixel in the hyperspectral images, the pixels of different classes in the adjacent pixels increase the spectral shift of the central pixel, while the pixels of the same class in the adjacent pixels help to reduce the spectral shift of the central pixel [29]. Inspired by reference [29], this paper proposes a weighted spatial-spectral kernel-collaborative representation method with Tikhonov regularization to mine spatial-spectral features for land cover classification, instead of directly averaging all the pixels (central pixels) and its corresponding spatial neighborhood pixels.

In addition, machine learning algorithms, especially deep learning, usually need sufficient labeled training samples to establish hyperspectral land cover classification models, so as to enhance the robustness of classification models. However, in practical hyperspectral applications, it is very difficult to label pixels, which usually consumes a lot of manpower and time [30]. Therefore, the lack of labeled samples is a great challenge in hyperspectral image classification [31]. To solve this problem, this paper attempts to use the proposed method to establish an effective land cover classification model in the case of small-size labeled samples.

The main contributions of this paper are as follows:

(1) A correlation coefficient-weighted spatial filtering operation is proposed to mine spatial-spectral features, which effectively reduces the spectral shift of the reconstructed central pixel.

(2) By introducing a weighted spatial filtering operation into the KCRT and DKCRT methods, weighted spatial-spectral KCRT (WSSKCRT) and weighted spatial-spectral DKCRT (WSSDKCRT) methods, respectively, are proposed for land cover classification.

(3) By optimizing parameters, the proposed method can effectively classify land cover types using hyperspectral images in the case of small-size labeled samples.

## 2. Materials and Methods

### 2.1. Data Collection

The hyperspectral scene in the experiment was collected by a Reflective Optics Spectrographic Imaging System (ROSIS) sensor mounted on a flight platform over the Pavia University in northern Italy. The spatial size of this scene is 610 × 340 pixels, with a high spatial resolution of 1.3 m. Additionally, this scene consists of 115 spectral bands. By removing 12 bands with high noises and water absorption, the remaining 103 bands, ranging from 0.43 to 0.86 µm, are used for the establishment and analysis of land cover classification models. In addition, there are nine land cover types with 42,776 labeled pixels in this hyperspectral scene, including *Asphalt*, *Meadows*, *Gravel*, *Trees*, *Paintedmetal sheets*, *Bare Soil*, *Bitumen*, *Self-Blocking Bricks*, and *Shadows*. The false-color image and ground truth of the acquired hyperspectral scene are shown in Figure 3a,b, respectively.

In hyperspectral scenes, each pixel represents a sample of one class. To satisfy the situation of small-size labeled samples, 60 labeled pixels in each class are randomly selected as training samples and the remaining pixels are as test samples. The specific division of samples is shown in Table 1.

**Table 1.** Land cover classes and division of samples in the hyperspectral scene.

| No. | Class | Total Samples | Training Samples | Test Samples |
|---|---|---|---|---|
| 1 | *Asphalt* | 6631 | 60 | 6571 |
| 2 | *Meadows* | 18,649 | 60 | 18,589 |
| 3 | *Gravel* | 2099 | 60 | 2039 |
| 4 | *Trees* | 3064 | 60 | 3004 |
| 5 | *Painted metal sheets* | 1345 | 60 | 1285 |
| 6 | *Bare Soil* | 5029 | 60 | 4969 |
| 7 | *Bitumen* | 1330 | 60 | 1270 |
| 8 | *Self-Blocking Bricks* | 3682 | 60 | 3622 |
| 9 | *Shadows* | 947 | 60 | 887 |
| | All classes | 42,776 | 540 | 42,236 |

### 2.2. Classification Methods

#### 2.2.1. Principle of the Original KCRT Method

Suppose a hyperspectral scene contains $C$ classes with $N$ labeled training samples, and the training samples can be expressed as $\mathbf{X} = [x_1, x_2, \cdots, x_N] \in \mathbb{R}^{d \times N}$, where $d$ is the dimension of hyperspectral data (i.e., the number of hyperspectral bands). Additionally, the training set of the $l$th class ($l = 1, 2, \ldots, C$) is denoted as $\mathbf{X}_l = [x_{l,1}, x_{l,2}, \cdots, x_{l,N_l}] \in \mathbb{R}^{d \times N_l}$, where $N_l$ represents the number of the training samples in the $l$th class, i.e., $\sum_{l=1}^{C} N_l = N$.

The essential idea of KCRT is to map each sample to a kernel-induced high-dimensional feature space through a nonlinear mapping function $\Phi$, enhancing the class separability. Then, each mapped test samples $\Phi(y)$ are linearly represented using the dictionary constructed by the mapped training samples $\Phi(\mathbf{X})$, where $\Phi(y) \in \mathbb{R}^{D \times 1}$ and $\Phi(\mathbf{X}) = [\Phi(x_1), \Phi(x_2), \cdots, \Phi(x_N)] \in \mathbb{R}^{D \times N}$ (D >> d is the dimension of high dimensional feature space). According to the definition of kernel function [22,23], the inner product of any two samples mapped by nonlinear mapping function can be expressed as kernel function, and the kernel function must satisfy Mercer's conditions. The kernel function used in KCRT is the Gaussian radial basis function (RBF). Therefore, the above statements can be expressed as follows:

$$k(x_i, x_j) = \Phi(x_i)^T \Phi(x_j) = \exp(-\gamma \|x_i - x_j\|_2^2) \tag{1}$$

where $\gamma$ $(\gamma > 0)$ is a parameter controlling the width of RBF. For KCRT, $\gamma$ is set as the median value of $1/(\|x_i - \bar{x}\|_2^2)$ ($i = 1, 2, \ldots, N$), where $\bar{x}$ is the mean value of all available training samples, i.e., $\bar{x} = ((\sum_{i=1}^{N} x_i)/N)$.

The representation coefficient vector $\boldsymbol{\alpha}$ in the kernel feature space is solved by $\ell_2$-norm regularization, i.e.,

$$\boldsymbol{\alpha} = \arg \min_{\boldsymbol{\alpha}^*} \|\Phi(y) - \Phi(\mathbf{X})\boldsymbol{\alpha}^*\|_2^2 + \lambda \|\boldsymbol{\Gamma}_{\Phi(y)}\boldsymbol{\alpha}^*\|_2^2 \tag{2}$$

where $\lambda$ is a global regularization parameter which is used to balance the minimization between the residual part and the regularization term, and the Tikhonov regularization term $\boldsymbol{\Gamma}_{\Phi(y)}$ in the kernel feature space can be expressed as

$$\boldsymbol{\Gamma}_{\Phi(y)} = \begin{bmatrix} \|\Phi(y) - \Phi(x_1)\|_2 & \cdots & 0 \\ \vdots & \ddots & \vdots \\ 0 & \cdots & \|\Phi(y) - \Phi(x_N)\|_2 \end{bmatrix} \tag{3}$$

where $\|\Phi(y) - \Phi(x_i)\|_2 = [k(y, y) + k(x_i, x_i) - 2k(y, x_i)]^{1/2}$, and $i = 1, 2, \ldots, N$. Then, the representation coefficient vector $\boldsymbol{\alpha}$ can be calculated with a closed-form solution as follows:

$$\boldsymbol{\alpha} = (\mathbf{K} + \lambda \boldsymbol{\Gamma}_{\Phi(y)}^T \boldsymbol{\Gamma}_{\Phi(y)})^{-1} k(\mathbf{X}, y) \tag{4}$$

where $\mathbf{K} = \Phi(\mathbf{X})^T \Phi(\mathbf{X}) \in \mathbb{R}^{N \times N}$ denotes the Gram matrix composed of $\mathbf{K}_{i,j} = k(x_i, x_j)$ ($i, j = 1, 2, \ldots, N$), and $k(\mathbf{X}, y) = [k(x_1, y), k(x_2, y), \cdots, k(x_N, y)]^T \in \mathbb{R}^{N \times 1}$. Finally, the obtained representation coefficient vector $\boldsymbol{\alpha} \in \mathbb{R}^{N \times 1}$ is divided into $C$ class-specific representation coefficient vectors according to the label information in the training set, i.e., $\boldsymbol{\alpha} = [\boldsymbol{\alpha}_1^T, \boldsymbol{\alpha}_2^T, \cdots, \boldsymbol{\alpha}_C^T]^T$. The mapped class-specific training samples $\Phi(\mathbf{X}_l)$ and the corresponding representation coefficient vector $\boldsymbol{\alpha}_l$ are used to reconstruct the test sample $y$, and the class with the minimal reconstruction error is attributed to $y$, whichis

$$\begin{aligned} \text{class}(\, y \,) &= \arg \min_{l=1,\cdots,C} \|\Phi(\mathbf{X}_l)\boldsymbol{\alpha}_l - \Phi(y)\|_2 \\ &= \arg \min_{l=1,\cdots,C} \sqrt{k(y, y) + \boldsymbol{\alpha}_l^T \mathbf{K}_l \boldsymbol{\alpha}_l - 2\boldsymbol{\alpha}_l^T k(\mathbf{X}_l, y)} \end{aligned} \tag{5}$$

where $\Phi(\mathbf{X}_l) = [\Phi(\mathbf{x}_{l,1}), \Phi(\mathbf{x}_{l,2}), \cdots, \Phi(\mathbf{x}_{l,N_l})]$ represents kernel sub-dictionary constructed by the training samples in the *l*th class, $\mathbf{K}_l = \Phi(\mathbf{X}_l)^T \Phi(\mathbf{X}_l) \in \mathbb{R}^{N_l \times N_l}$ denotes the Gram matrix composed of the training samples in the *l*th class, and $k(\mathbf{X}_l, \mathbf{y}) = [k(\mathbf{x}_{l,1}, \mathbf{y}), k(\mathbf{x}_{l,2}, \mathbf{y}), \cdots, k(\mathbf{x}_{l,N_l}, \mathbf{y})]^T \in \mathbb{R}^{N_l \times 1}$.

### 2.2.2. Principle of the Original DKCRT Method

On the basis of the KCRT method, the DKCRT method adds a discriminative regularization term to consider the correlation among different classes, so as to enhance the class separability. The optimization problem of the representation coefficient vector $\boldsymbol{\alpha}$ in the kernel feature space can be mathematically reduced to the following form

$$\boldsymbol{\alpha} = \arg \min_{\boldsymbol{\alpha}^*} \|\Phi(\mathbf{y}) - \Phi(\mathbf{X})\boldsymbol{\alpha}^*\|_2^2 + \lambda\|\boldsymbol{\Gamma}_{\Phi(\mathbf{y})}\boldsymbol{\alpha}^*\|_2^2 + \beta\sum_{i=1}^{C}\sum_{j=1}^{C}(\Phi(\mathbf{X}_i)\boldsymbol{\alpha}_i^*)^T(\Phi(\mathbf{X}_j)\boldsymbol{\alpha}_j^*) \tag{6}$$

where $\beta$ is a positive regularization parameter controlling the contribution of the discriminative regularization term. The closed-form solution of the representation coefficient vector $\boldsymbol{\alpha}$ can be expressed as

$$\boldsymbol{\alpha} = [(1+\beta)\mathbf{K} + \lambda\boldsymbol{\Gamma}_{\Phi(\mathbf{y})}^T\boldsymbol{\Gamma}_{\Phi(\mathbf{y}} + \beta\mathbf{Q}]^{-1}k(\mathbf{X}, \mathbf{y}) \tag{7}$$

where $\mathbf{Q}$ is expressed in the following form:

$$\mathbf{Q} = \begin{bmatrix} \mathbf{K}_1 & \cdots & 0 \\ \vdots & \ddots & \vdots \\ 0 & \cdots & \mathbf{K}_C \end{bmatrix} \tag{8}$$

As with the KCRT method, the test sample $\mathbf{y}$ is classified using formula (5).

### 2.2.3. Principle of the Original KCRT-CK and JDKCRT Method

KCRT and DKCRT only use spectral features in hyperspectral images to classify land cover types, while KCRT-CK and JDKCRT combine spatial and spectral features to improve the classification accuracy for land cover types. Both KCRT-CK and JDKCRT mine spatial-spectral features using a spatial filtering operation that directly averages all the pixels (central pixels) and its corresponding spatial neighborhood pixels in hyperspectral images. The specific mathematical expression is as follows:

Suppose that $\Psi = \{\mathbf{x}_{0,0}, \mathbf{x}_{0,1}, \ldots, \mathbf{x}_{0,n \times n-1}\}$ is the spatial neighborhood pixel set corresponding to the central pixel $\mathbf{x}_{0,0}$ under the window of $n \times n$. Note that $\Psi$ contains the central pixel $\mathbf{x}_{0,0}$ itself. The mean value of all samples in $\Psi$ can be calculated as follows:

$$\hat{\mathbf{x}}_0 = \frac{1}{n \times n}\sum_{i=0}^{n \times n-1} \mathbf{x}_{0,i} \tag{9}$$

where $\hat{\mathbf{x}}_0$ is the reconstructed central pixel (i.e., spatial-spectral features of $\mathbf{x}_{0,0}$). In this way, all pixels in the hyperspectral scene are traversed. After that, the KCRT and DKCRT methods are used to classify the land cover types in hyperspectral scene.

### 2.2.4. Principle of the Proposed WSSKCRT and WSSDKCRT Method

It has been introduced in the Introduction part that it usually includes some pixels of different classes from the central pixel in its spatial adjacent pixels, and the pixels of different classes in the spatial adjacent pixels increase the spectral shift of the central pixel. Therefore, the reconstructed central pixel (i.e., spatial-spectral features) by directly averaging adjacent pixels contains a large amount of noise caused by spectral shift. To solve this problem, a correlation coefficient-weighted spatial filtering operation is proposed in this paper, and the proposed spatial filtering operation is introduced into KCRT and DKCRT methods, denoted as WSSKCRT and WSSDKCRT, respectively. The specific formulas are expressed as follows:

Similarly, suppose the spatial neighborhood pixel set of a central pixel $x_{0,0}$ under the window of $n \times n$ is $\Psi = \{x_{0,0}, x_{0,1}, \dots, x_{0,n \times n-1}\}$. Firstly, the correlation coefficient between each pixel in the spatial neighborhood pixel set $\Psi$ and the central pixel is calculated, respectively. The results can be expressed as $R = \{r_{0,0}, r_{0,1}, \dots, r_{0,n \times n-1}\}$, where $r_{0,0}$ represents the correlation coefficient of the central pixel itself (i.e., 1), and $r_{0,i}$ represents the correlation coefficient between the central pixel $x_{0,0}$ and the spatial neighborhood pixel $x_{0,i}$ $(i = 1,2, \dots, n \times n-1)$. The larger the absolute value of correlation coefficient is between the spatial neighborhood pixel $x_{0,i}$ and the central pixel $x_{0,0}$, the higher the probability is that it belongs to the same class as the central pixel, so the corresponding spatial neighborhood pixel $x_{0,i}$ should be assigned a larger weight. To prevent the influence of negative correlation value on the weight, the absolute value of the obtained correlation coefficient is normalized to 0–1, which is used as the weight value of the corresponding spatial neighborhood pixel, i.e.,

$$w_{0,i} = \frac{|r_{0,i}|}{\sum_{i=0}^{n \times n-1} |r_{0,i}|} \tag{10}$$

The reconstructed central pixel $\hat{x}_0$ can be calculated as follows:

$$\hat{x}_0 = \sum_{i=0}^{n \times n-1} w_{0,i} x_{0,i} \tag{11}$$

All the pixels in the hyperspectral scene are traversed using this weighted spatial filtering operation to mine spatial-spectral features. Finally, land cover types in hyperspectral scene are classified by the KCRT and DKCRT methods.

## 3. Results and Discussion

### 3.1. Hyperspectral Data Preprocessing

In one hyperspectral scene, the spectral curves of the same ground object usually produce amplitude shift due to different geometrical structure and smoothness, which will degrade the performance of classification model [29]. To illustrate amplitude shift, the pixels of three kinds of land cover types, i.e., *Asphalt*, *Gravel*, and *Bare Soil*, are selected in the acquired hyperspectral scene, and 60 pixels are randomly selected in each kind of land cover type. The spectral response curves are shown in Figure 1 a–c. It can be seen from the figure that the shape and trend of spectral curves of the same ground object is basically invariant, but the reflectance level (i.e., spectral amplitude) is obviously different, which is the amplitude shift mentioned above. To alleviate the amplitude shift, the amplitude normalization (AN) method proposed in reference [29] is used to preprocess the original hyperspectral data in this paper. The principle of AN is as follows:

Assuming that $x_i$ is a pixel of one class in the hyperspectral scene, the amplitude of $x_i$ is normalized by the following formula:

$$\hat{x}_i = \frac{x_i}{\sum_{b=1}^{d} |x_{ib}|} \tag{12}$$

where $\hat{x}_i$ is the pixel preprocessed by AN, $x_{ib}$ represents the reflectance of the $b$th band, and $d$ represents the number of bands. The amplitude of all ground object pixels in the acquired hyperspectral scene is normalized using formula (12). It can be seen from Figure 1d–f that the spectral curves of the same ground object become compact and concentrated after AN pretreatment., which indicates the amplitude shift of spectral curves of the same ground object is effectively alleviated. Therefore, the spectral data preprocessed by the AN method is used for subsequent modeling analysis.

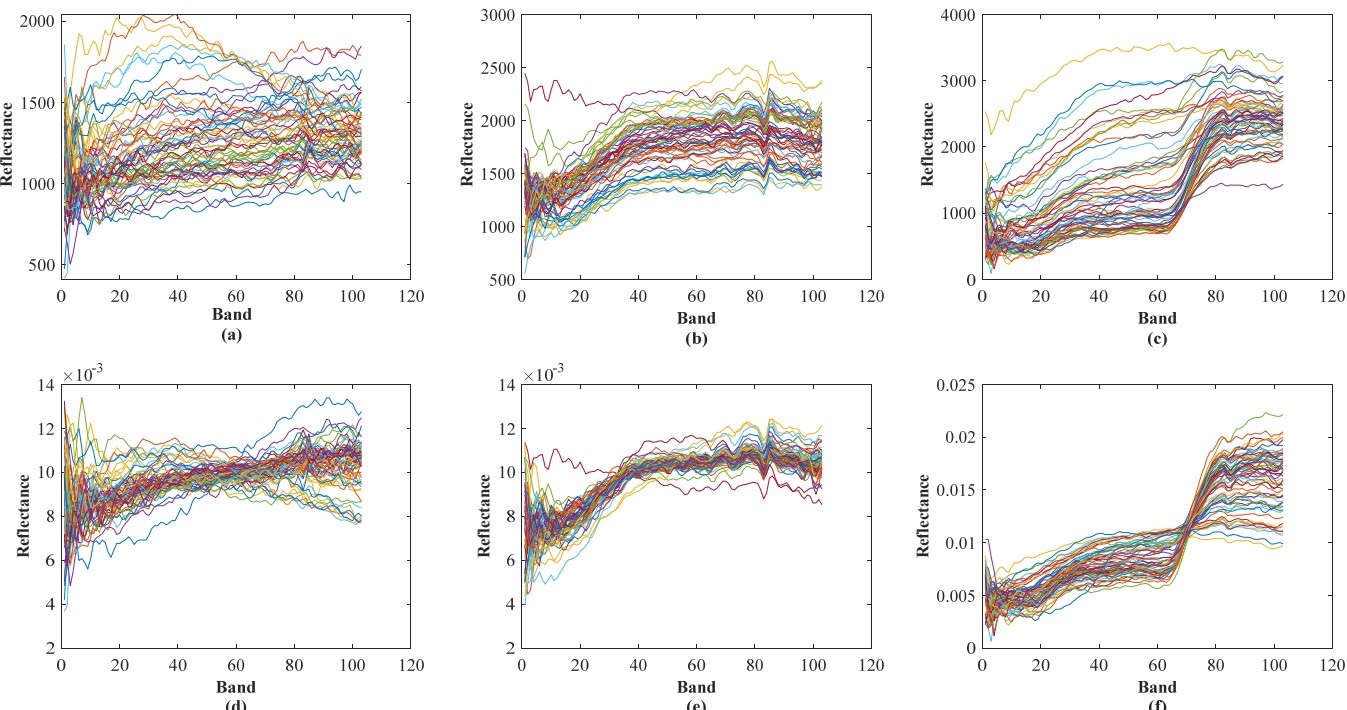

**Figure 1.** Original spectral response of (**a**) *Asphalt*, (**b**) *Gravel*, and (**c**) *Bare Soil*, and pretreated spectral response of (**d**) *Asphalt*, (**e**) *Gravel*, (**f**) *Bare Soil* with the AN method.

### 3.2. Parameter Optimization

To verify the effectiveness of the proposed WSSKCRT and WSSDKCRT methods for land cover classification, the classification performance is compared with that of the KCRT, DKCRT, KCRT-CK, and JDKCRT methods. In addition, to ensure the fairness of the experiment, the classification performance of all methods is compared under the corresponding optimal parameters.

There are three main parameters (i.e., $\lambda$, $\beta$, and spatial filtering window size $T$) that produce a significant impact on the classification performance of the above-mentioned methods, in which $\lambda$ is a main parameter for KCRT, $\lambda$ and $\beta$ are main parameters for DKCRT, $\lambda$ and $T$ are main parameters for KCRT-CK and WSSKCRT, $\lambda$, $\beta$ and $T$ are main parameters for JDCRT and WSSDKCRT, and these parameters need to be optimized, respectively. In this paper, the corresponding parameters of each method are optimized using 540 labeled training samples randomly selected in the hyperspectral scene and a five-fold cross-validation strategy. During the optimization process, $\lambda$ and $\beta$ are chosen from the given intervals $\{10^{-7}, 10^{-6}, 10^{-5}, 10^{-4}, 10^{-3}, 10^{-2}, 10^{-1}, 1\}$, and window size $T$ is chosen from the given intervals $\{3 \times 3, 5 \times 5, 7 \times 7, 9 \times 9, 11 \times 11\}$. The classification performance for each method under different parameters is shown in Figure 2. For JDCRT and WSSDKCRT, there are three parameters (i.e., $\lambda$, $\beta$ and $T$) to be optimized at the same time, thus using the surface of different colors to represent the corresponding window size $T$, as shown in Figure 2c,e. In addition, an asterisk (*) is used to represent the position of the optimal parameters for each method in the three-dimensional graph. The optimal parameter settings for each method are shown in Table 2.

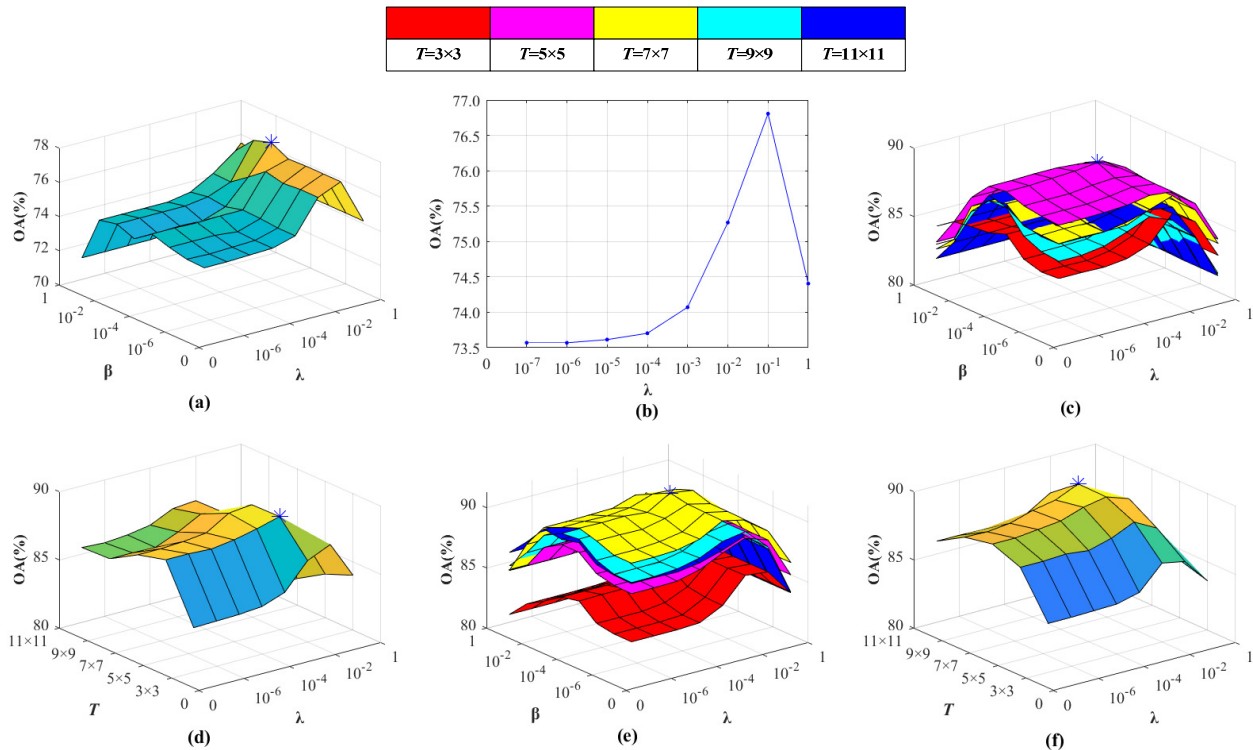

**Figure 2.** Classification performance for (**a**) DKCRT, (**b**) KCRT, (**c**) JDKCRT, (**d**)KCRT-CK, (**e**) WSSD-KCRT, and (**f**) WSSKCRT under different parameters.

**Table 2.** Optimal parameter settings for each method.

| Parameters | Methods | | | | | |
|---|---|---|---|---|---|---|
| | **DKCRT** | **KCRT** | **JDKCRT** | **KCRT-CK** | **WSSDKCRT** | **WSSKCRT** |
| $\lambda$ | $10^{-1}$ | $10^{-1}$ | $10^{-3}$ | $10^{-2}$ | $10^{-3}$ | $10^{-2}$ |
| $\beta$ | $10^{-3}$ | No application | $10^{-4}$ | No application | $10^{-4}$ | No application |
| $T$ | No application | No application | $5 \times 5$ | $5 \times 5$ | $7 \times 7$ | $9 \times 9$ |

### 3.3. Land Cover Classification

The above-mentioned methods classify the land cover types in the acquired hyperspectral scene under the corresponding optimal parameters. Additionally, individual class accuracy, overall accuracy (OA), average accuracy (AA), and kappa statistic (Kappa) are employed to evaluate the classification performance of each method. To avoid random error and any bias, each method is conducted repeatedly for 10 runs. Additionally, in each run, 60 pixels per class are randomly selected as training samples and the remaining pixels are taken as test samples. The average value of the results of these 10 runs is taken as the final classification accuracy. The classification results of land cover types by each method are shown in Table 3 and Figure 3. The best classification results are presented in highlighting font in Table 3.

**Table 3.** Classification accuracy for land cover types.

| Class | DKCRT | KCRT | JDKCRT | KCRT-CK | WSSDKCRT | WSSKCRT |
|---|---|---|---|---|---|---|
| *Asphalt* | 74.47 | 71.71 | **92.34** | 92.22 | 91.54 | 91.56 |
| *Meadows* | 81.45 | 80.59 | 95.41 | 95.22 | 96.70 | **97.51** |
| *Gravel* | 85.85 | 77.78 | 94.77 | 90.32 | **95.70** | 91.48 |
| *Trees* | 94.00 | 94.41 | 96.32 | 96.43 | **96.81** | 96.66 |
| *Painted metal sheets* | 99.57 | 99.44 | 99.98 | **100.00** | 99.70 | 99.65 |
| *Bare Soil* | 80.56 | 78.03 | 94.01 | 94.07 | 96.27 | **97.13** |
| *Bitumen* | 92.61 | 90.86 | 98.38 | 96.83 | **99.35** | 97.50 |
| *Self-Blocking Bricks* | 61.65 | 77.03 | 69.80 | 88.21 | 73.34 | **90.92** |
| *Shadows* | 97.42 | 97.96 | 99.53 | **99.71** | 98.65 | 97.68 |
| OA (%) | 80.89 | 80.70 | 92.92 | 94.16 | 94.02 | **95.69** |
| AA (%) | 85.29 | 85.31 | 93.39 | 94.78 | 94.23 | **95.56** |
| Kappa | 0.7535 | 0.7512 | 0.9064 | 0.9228 | 0.9208 | **0.9429** |

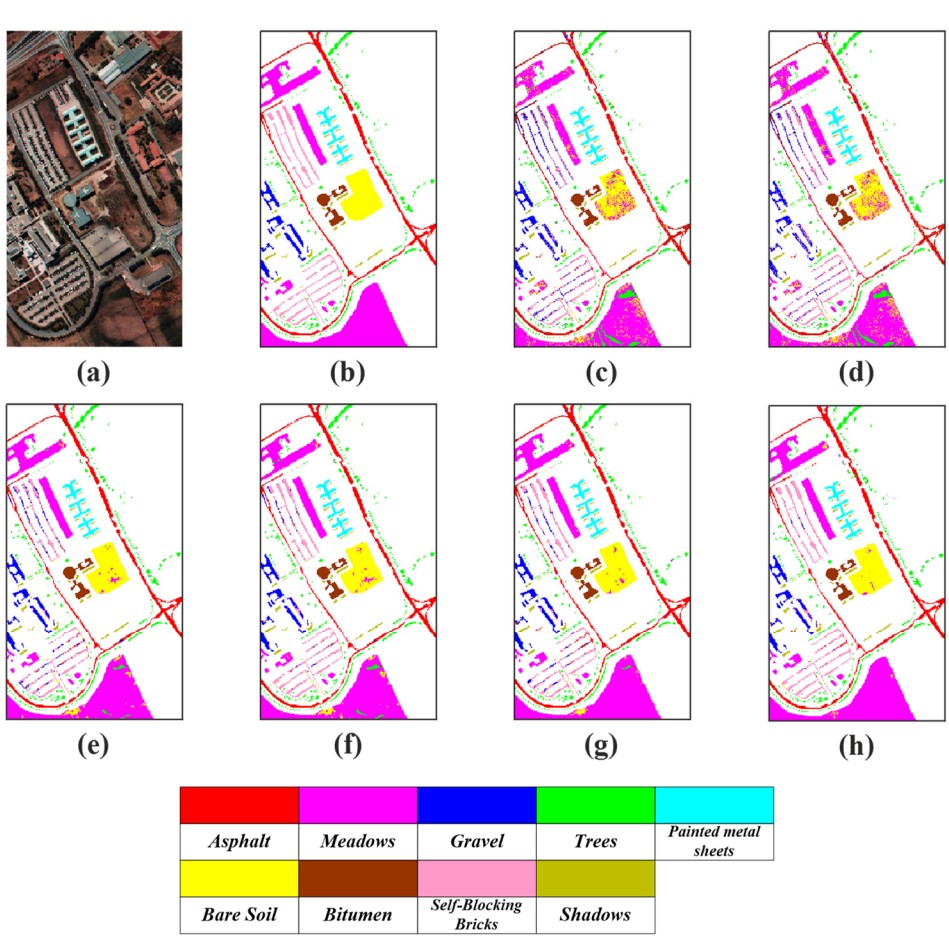

**Figure 3.** (**a**) False-color image, (**b**) ground truth, and land cover classification maps generated by (**c**) DKCRT, (**d**) KCRT, (**e**) JDKCRT, (**f**) KCRT-CK, (**g**) WSSDKCRT, and (**h**) WSSKCRT.

It can be seen from Table 3 that the proposed WSSKCRT method achieves the best classification performance, in which OA, AA, and Kappa for land cover classification is 95.69%, 95.56%, and 0.9429, respectively. Additionally, there is the least classification noise in the classification map obtained by WSSKCRT as shown in Figure 3h. Moreover, the classification performance of WSSKCRT and WSSDKCRT is better than that of KCRT-CK and JDKCRT, respectively, which indicates that the proposed weighted spatial filtering operation can effectively alleviate the spectral shift caused by adjacency effect when mining

the spatial-spectral features of hyperspectral images. Compared with other methods, DKCRT and KCRT possess the worst classification performance, due to not considering the spatial features of hyperspectral images, and there is more classification noise in the classification maps obtained by DKCRT and KCRT as shown in Figure 3c,d. In addition, all methods utilize only 540 labeled training samples (60 training samples per class) to establish the land cover classification models and classify the remaining 42,236 ground object samples. In this case, the proposed WSSKCRT and WSSDKCRT methods achieve the promising classification performance with the OA over 94%, which indicates that the proposed methods can effectively classify land cover types under the situation of small-size labeled samples.

## 4. Conclusions

In this paper, land cover types are classified by using hyperspectral images and the kernel collaborative representation method. The conclusions of this paper are summarized as follows:

(1) The proposed WSSKCRT method achieves the best classification result, in which OA, AA, and Kappa is 95.69%, 95.56%, and 0.9429, respectively.

(2) WSSKCRT and WSSDKCRT outperform KCRT-CK and JDKCRT, respectively, which indicates that the proposed weighted spatial filtering operation can effectively alleviate the spectral shift caused by adjacency effect when mining the spatial-spectral features of hyperspectral images.

(3) WSSKCRT and WSSDKCRT methods obtain the OA over 94% with only 540 labeled training samples, which indicates that the proposed methods can effectively classify land cover types under the situation of small-size labeled samples.

The experimental results show that the proposed WSSKCRT and WSSDKCRT methods can effectively alleviate the spectral shift caused by adjacency effect, and can effectively classify land cover types under the situation of small-size labeled samples. However, like the traditional collaborative representation methods, the WSSKCRT and WSSDKCRT methods utilize the labeled training samples of all classes to construct a dictionary to represent and classify each test sample, which may degrade the classification performance of collaborative representation models to some extent, due to the irrelevant classes to test samples. In the follow-up research, we will focus on exploring the appropriate nearest neighbor collaborative representation mechanism, that is, using the classes nearest to each test sample to represent and classify a corresponding test sample, so as to eliminate irrelevant classes and further improve the classification performance of collaborative representation models. In addition, the proposed methods achieve effective classification of land types in a hyperspectral scene, with the same spatial resolution and a relatively small size. The classification performance of the proposed methods for hyperspectral scenes with different spatial resolution and larger region needs to be further analyzed and studied.

**Author Contributions:** Data curation, R.Y.; Methodology, R.Y.; Supervision, Y.W. and Q.Z.; Validation, B.F. and R.W.; Writing—original draft, R.Y.; Writing—review & editing, B.F. All authors have read and agreed to the published version of the manuscript.

**Funding:** This work was funded by the Basic Research Fund of Agricultural Information Institute of CAAS (Grant No. JBYW-AII-2021-02) and the Basic Research Fund of Chinese Academy of Agricultural Sciences, China (Grant No. Y2020YJ18).

**Institutional Review Board Statement:** Not applicable.

**Informed Consent Statement:** Not applicable.

**Data Availability Statement:** Hyperspectral data set of Pavia University can be obtained from http://www.ehu.eus/ccwintco/index.php?title=Hyperspectral_Remote_Sensing_Scenes.

**Acknowledgments:** We acknowledge Paolo Gamba from Pavia University for providing the ROSIS hyperspectral data of Pavia University for the research of land cover classification.

**Conflicts of Interest:** The authors declare no conflict of interest.

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
