# Peer review of "Land Cover Classification from Hyperspectral Images via Weighted Spatial-Spectral Kernel Collaborative Representation with Tikhonov Regularization"

_land, doi:10.3390/land11020263_

Round 1
Reviewer 1 Report
Review
For classification of land cover, vegetation cover, etc. different researchers use different remote sensing data and classification methods. The accuracy of the classification of objects on Earth, and therefore the reliable display of the study area during mapping, depends on the spatial resolution of satellite images and the correctly selected method.
In this paper, the authors present a land cover classification using hyperspectral images and the kernel collaborative representation method. The researchers made a comparative analysis of 6 methods in the classification of 9 types of land cover, as a result of which a method was identified that provides the best classification result, and the advantages and disadvantages of other methods are presented. The conclusions and recommendations of the authors based on the results of the study, in my opinion, have a rather important scientific and practical significance.
The article is well structured, the authors have done a fairly in-depth analysis, the conclusions are logical, in general, the study is presented quite well and can be recommended for publication.

Author Response
Dear Reviewer,
Please see the attachment about our reply to the comments.

Reviewer 2 Report
Summary:
This manuscript provides an excellent example of Land Cover Classification from Hyperspectral Images via Weighted Spatial-Spectral Kernel Collaborative Representation with Tikhonov Regularization. The science and methodology of the manuscript appear sound, and adequately cited. Pretty well written paper, so I do not have many major comments. One point that could perhaps be strengthened is more of an indication to the readers of the level of uniqueness of the study comparing with previous studies if there is any. I believe a minor level of revisions should be made to the paper before it is ready to be considered for publication with Land.
General Comment:
In conclusions, some discussion should include, e.g. the advantages of WSSKCRT and WSSDKCRT methods, and its limitations.
In practically, the size of study area is small, only 610×340 pixels with a high 119 spatial resolution of 1.3 m, how does the WSSKCRT and WSSDKCRT methods work for bigger area? Moreover, How the different spatial resolution input data impact on the mapping accuracy if you have done any such analysis?
Author Response
Dear Reviewer,
Please see the attachment about our reply to the comment.
